# Inhalation Therapy with Nebulized Capsaicin in a Patient with Oropharyngeal Dysphagia Post Stroke: A Clinical Case Report

**DOI:** 10.3390/geriatrics7020027

**Published:** 2022-02-28

**Authors:** Anna Maria Pekacka-Egli, Jana Herrmann, Marc Spielmanns, Arthur Goerg, Katharina Schulz, Eveline Zenker, Wolfram Windisch, Stefan Tino Kulnik

**Affiliations:** 1Department for Pulmonary Medicine and Sleep Medicine, Zürcher RehaZentren, Klinik Wald, 8636 Wald, Switzerland; jana.herrmann@zhreha.ch (J.H.); marc.spielmanns@zhreha.ch (M.S.); arthur.georg@zhreha.ch (A.G.); katharina.schulz@zhreha.ch (K.S.); eveline.zenker@zhreha.ch (E.Z.); 2Department for Neurology and Neurorehabilitation, Zürcher RehaZentren, Klinik Wald, 8636 Wald, Switzerland; 3Department for Pulmonary Medicine, Faculty of Health, University Witten-Herdecke, 58455 Witten, Germany; windisch.w@kliniken-koeln.de; 4Faculty of Health, Social Care and Education, Kingston University and St. George’s University of London, London SW17 0RE, UK; ku70991@kingston.ac.uk

**Keywords:** capsaicin, cough, dysphagia, nebulization, pneumonia, rehabilitation, respiratory physiotherapy, speech and language therapy, stroke, case report

## Abstract

Dysphagia and aspiration risk are common sequelae of stroke, leading to increased risk of stroke-associated pneumonia. This is often aggravated by stroke-related impairment of cough, the most immediate mechanical defense mechanism against aspiration. In humans, reflex cough can be repeatedly and safely elicited by inhalation of nebulized capsaicin, a compound contained in chili peppers. Could this cough-eliciting property of capsaicin support the recovery of stroke survivors who present with dysphagia and aspiration risk? We present a clinical case report of a 73-year-old man, admitted to inpatient stroke rehabilitation following a right middle cerebral artery infarct with subsequent dysphagia and hospital-acquired pneumonia. A course of daily inhalation therapy with nebulized capsaicin was initiated, triggering reflex coughs to support secretion clearance and prevent recurrence of pneumonia. Clinical observations in each inhalation therapy session demonstrate good patient response, safety and tolerability of nebulized capsaicin in this mode of application. Repeated Fiberoptic Endoscopic Evaluation of Swallowing (FEES) assessments show concurrent improvement in the patient’s swallowing status. Inhalation therapy with nebulized capsaicin may offer a viable treatment to facilitate coughing and clearing of secretions, and to minimize aspiration and risk of aspiration-related pneumonia post stroke. Further investigation in a randomized controlled trial design is warranted.

## 1. Introduction

Stroke is considered the second leading cause of death worldwide and is the most common cause of severe disability after the age of 40 [1,2]. A common consequence of stroke is oropharyngeal dysphagia, the dysfunction of one or more parts of the swallowing apparatus including the mouth, tongue, oral cavity, pharynx, airway, and esophagus [3]. Reported incidence rates for post-stroke dysphagia range from 8% to 80%, depending on the definition of dysphagia and the method of its detection [4].

A common complication of dysphagia is the aspiration of food, drink, saliva or stomach contents into the lower airways, leading to increased risk of stroke-associated pneumonia (SAP) [5,6]. It has been reported that dysphagic stroke survivors are at an approximately three-fold risk of developing pneumonia compared to non-dysphagic stroke survivors, and those with confirmed aspiration exhibit an eleven-fold risk compared to non-aspirators [7]. Early dysphagia screening and clinical swallowing assessment followed by appropriate clinical management strategies can reduce aspiration and subsequently lower the risk of pneumonia post stroke [8]. It is important to note, that often screening is focused only on dysphagia without consideration of aspiration and vice-versa. This could be misleading, because dysphagia can occur without aspiration [9]. Conversely, high sensitivity bed site evaluations (BSE) also designed to detect aspiration and tested against FEES are more likely to depict the real situation, thus being more useful to design studies on post-stroke aspiration prevention [10,11]. The presence of silent aspiration presents a further difficulty, as most BSEs fail to detect silent aspiration, while the presence of undiagnosed silent aspiration leads to an increase in the relative risk of pneumonia and poorer stroke clinical outcomes.

Recent studies have reported higher stroke-associated pneumonia in stroke patients who failed high-sensitivity screening for dysphagia compared to those who passed the screening [12]. Moreover, pneumonia incidence was higher in stroke patients who passed low-sensitivity screening for dysphagia compared to those who passed high-sensitivity BSEs which could also detect silent aspiration [13].

In the presence of post-stroke dysphagia and aspiration risk, the likelihood of developing pneumonia is also influenced by the integrity of mechanical defense mechanisms against aspiration, i.e., throat clearing and cough [14]. Cough is the most immediate and important mechanical defense mechanism against aspiration [15]. A number of studies have shown that stroke can lead to significant impairment in both volitional and reflex cough [16,17], and that decreased cough flow and impaired reflex cough sensitivity are associated with a higher incidence of pneumonia post stroke [17,18,19].

In humans, reflex cough may be elicited by inhalation of nebulized capsaicin, a pungent molecule contained in chili peppers with the formula C_18_H_27_NO_3_ [20]. Above a certain threshold, inhaled nebulized capsaicin will trigger reflex coughs by acting on chemoreceptors in the lower airways [21], a property of capsaicin that has been used for reflex cough sensitivity testing (inhalation cough challenge) [22,23] in respiratory research since the 1980s [24], with a documented excellent safety record in healthy volunteers as well as in patients with asthma, chronic obstructive pulmonary disease (COPD), pathologic cough, and other respiratory diseases [25].

There is currently no clinically established therapeutic application of inhaled nebulized capsaicin. But could it be possible to utilize the cough-eliciting property of capsaicin for the treatment of stroke survivors who present with dysphagia and aspiration risk? Repeated administration of inhalation therapy with nebulized capsaicin may benefit individuals with post-stroke dysphagia and aspiration risk, by aiding clearance of residue from the laryngopharynx and lower airways through strong involuntary cough maneuvers. Moreover, the activation of central neural cough pathways by repeated triggering of reflex coughs through capsaicin may potentially also reinforce and support recovery of the central regulation of cough, post stroke. Incorporated into post-stroke care and rehabilitation, inhalation therapy with nebulized capsaicin may thereby contribute to dysphagia management and the prevention of aspiration-related pneumonia.

In this clinical case report, we present the application of a course of inhalation therapy with nebulized capsaicin during inpatient rehabilitation of a patient with oropharyngeal dysphagia and high aspiration risk post stroke. We discuss our clinical reasoning and scientific rationale for this treatment in this case, and we offer future perspectives for inhalation therapy with nebulized capsaicin in stroke care and rehabilitation.

## 2. Case Presentation

A 73-year-old man was admitted to our inpatient rehabilitation center for continued multidisciplinary rehabilitation post stroke. A timeline of the patient’s entire episode of care is provided in Figure 1. 

In his initial presentation, the patient had been admitted to a District General Hospital for investigation of cardiac complaints in July of 2021. Two days later, he had developed new weakness of the left side of the body and speech disorder. He was diagnosed with an ischemic insult of the media flow area of the right cerebral hemisphere. Due to a reduced level of alertness, the patient was transferred to the Acute Stroke Unit (ASU) of the University Hospital for monitoring and consideration of a craniectomy. He received multidisciplinary acute stroke management without craniectomy and with temporary oral and intravenous anticoagulation therapy. Thrombolysis was not carried out. No history of pneumonia was reported from the ASU.

Seven days post stroke onset, the patient was moved to a general medical ward, where he was treated for a further two weeks. Ten days post stroke, fever, worsening cough and dyspnea raised the suspicion of pneumonia. Subsequent chest X-ray imaging (Figure 2) showed incipient right-sided infiltrations in the lungs and ventilation disturbance in the left subfield with a washed-out lateral recess. A diagnosis of hospital acquired pneumonia with suspected aspiration was made in accordance with the Pneumonia in Stroke Consensus Group recommendations [26], based on symptoms of fever, worsening cough and dyspnea, crackles on auscultation, raised C-reactive protein (CRP) of 97.4, and new infiltrations on chest X-ray. A 7-day course of treatment with co-amoxicillin (2 g twice daily) was completed with good response, as seen in the CRP decrease to 24 after 4 days.

Three weeks post stroke onset, the patient was discharged from the acute hospital and transferred to our inpatient neurological rehabilitation center. The discharge report indicated the development of stroke-related impairment and disability (Table 1) and the following information relating to the dysphagia diagnosis and management at the time of discharge from the acute hospital: Severity of dysphagia according to Bogenhausener Dysphagie Score (BODS) [27]: 6 points (moderate severity of dysphagia);Modified diet, recommending modified food texture at International Dysphagia Diet Standardization Initiative (IDDSI) [28] level 4 and thickened drinks at IDDSI level 2;Nutritional intake had initially been managed via nasogastric tube. Weaning from tube feeding had been carried out prior to discharge from the acute hospital;Dysphagia management included therapy according to Facial-Oral Tract Therapy (FOTT) [29], modified diet and assistance with feeding.

## 3. Investigations

The patient was admitted to our inpatient rehabilitation center on 20 August 2021 and underwent initial nursing and medical assessments. The patient was then enrolled for multimodal multidisciplinary neurological rehabilitation, including physiotherapy, occupational therapy, neuropsychology and speech and language therapy (SLT). His Functional Independence Measure (FIM) [32] score at the beginning of rehabilitation was 37/91 (motor subscale), 12/35 (cognition subscale) and 49/126 points (total score).

### Initial Speech and Language Therapy Assessment

The initial SLT consultation at our inpatient rehabilitation center was conducted on 20 August 2021. Risk of dysphagia was initially reassessed with the Standardized Swallowing Assessment (SSA) [33] and was positive for aspiration predictors. Therefore, a Clinical Swallowing Evaluation (CSE) was completed later on the same day using the Neurogene Oropharyngale Dysphagie (NOD) Stufenkonzept [34], a standardized German version of CSE. The following problems were identified:Reduced tongue motor skills with tongue deviation to the left.Velum asymmetry (reduced tone on the left) with rhinolalia during speech production.Weak/ineffective voluntary cough.Ineffective voluntary throat clearing.Continuous wet voice without effective expectoration.Patient unaware of wet voice and ineffective expectoration.Negative modified (IDDSI 2) 3-oz water swallow test [35].Central facial nerve palsy (left) with residual oral branch weakness.Dysarthrophonia.

Based on these CSE results, the patient was scheduled for further assessment through Fiberoptic Endoscopic Evaluation of Swallowing (FEES). In the meantime, his dysphagia was managed by modified diet (food texture at IDDSI level 4, thickened drinks at IDDSI level 2) under mealtime supervision by nursing staff. Medication was administered in ground form mixed with apple sauce.

The patient underwent FEES on 25 August 2021. FEES was performed according to the recommendations by Langmore [36]. Ratings of the Murray Rating of Secretions scale (ROS) [37] and the Penetration Aspiration Scale (PAS) [38] were conducted according to international standards. We observed:ROS 3 (laryngeal penetrations of secretion).PAS 5 for thickened water (IDDSI 2).PAS 3 for apple sauce (IDDSI 4).Sensory testing [36]: no patient reaction to light touch of the scope to the epiglottis, no pharyngeal constriction.

## 4. Dysphagia Management and Rehabilitation

### 4.1. Dysphagia Management

Based on FEES results from 25 August 2021, the patient’s dysphagia management remained unchanged, i.e., modified diet (food texture at IDDSI level 4, thickened drinks at IDDSI level 2) under mealtime supervision by nursing staff.

### 4.2. Rehabilitation of Facial Weakness and Dysphagia

An SLT rehabilitation treatment plan was devised to address the patient’s left-sided facial weakness with resulting drooling, and dysphagia. SLT treatment was delivered in daily sessions (Monday through Friday, 30–45 min per session) during the patient’s entire rehabilitation stay. Drooling and facial weakness were treated with manually facilitated active movements (Proprioceptive Neuromuscular Facilitation [39]) and ultrasound therapy [40].

Dysphagia was treated with functional dysphagia therapy (FDT) [41], focusing on vocal fold closure, pharyngeal contraction, laryngeal elevation, enhancement of pharyngeal sensitivity through thermal-kinesthetic stimulation, and adjustment of clearing mechanisms. 

### 4.3. Monitoring for Aspiration

Monitoring screenings for aspiration were performed with a 3-oz water swallow test with thickened water (IDDSI 1) on 16 September 2021 and 21 September 2021. During these assessments, no clinical signs of aspiration were found. A repeat FEES examination was conducted on 16 September 2021 with unchanged findings from 25 August 2021. In combination with gradual improvement of drooling, the possibility of reducing the recommended thickening level for liquids to enhance the patient’s quality of life was considered. However, during a subsequent multidisciplinary case discussion on 24 September 2021, it was stated that the patient had been showing frequent signs of aspiration (coughing and ineffective throat clearing) after drinking about one hour after mealtimes. Another FEES was conducted on 28 September 2021 which showed:ROS 2 (up to 25% pooling of secretion in the valleculae and sinus piriformes).PAS 5 for thickened water (IDDSI 2).PAS 8 for thickened water (IDDSI 1).Sensory testing: no patient reaction to light touch of the scope to the epiglottis, minimal pharyngeal constriction.Highly inflamed/reddened mucous membrane and edema between arytenoid cartilage and vocal folds—possibly as a result of coughing and excessive throat clearing.

Based on these FEES results, indicating silent aspiration of thickened liquids (IDDSI 1) with inadequate clearing mechanisms (particularly the throat clearing), a decision was made to commence daily inhalation therapy with nebulized capsaicin from 1 October 2021.

## 5. Inhalation Therapy with Nebulized Capsaicin

### 5.1. Indication

At our inpatient rehabilitation center, inhalation therapy with nebulized capsaicin is at times considered to complement therapy for neurogenic dysphagia and when a patient presents with ROS >1, reduced response to sensory testing during FEES, reduced airway-protective reflexes and excessive secretions. The aim of nebulized capsaicin inhalation is to support clearance and protection of the lower airways by stimulating repeated reflexive coughs, thereby contributing to the prevention of aspiration-related pneumonia. 

Additional daily SLT sessions (Monday through Friday) were scheduled in the mornings for the delivery of capsaicin inhalation. These were followed by additional daily respiratory physiotherapy sessions (breathing exercises, patient education, mobilization). In parallel, the frequency and intensity of functional dysphagia therapy was reduced, so as to reduce overall therapy burden on the patient.

### 5.2. Preparation and Delivery

The capsaicin solution for nebulization at our rehabilitation center is prepared off-label by the in-house pharmacy (stock solution of 8 micromol/L capsaicin in normal saline). The stock solution is prepared fresh for daily use and then discarded. Immediately prior to delivering the inhalation therapy, the stock solution is filled into the receptacle of an air-driven nebulizer with face mask (Cirrus^TM^2 nebulizer with Intersurgical EcoLite^TM^ mask kit, Intersurgical GmbH, (Germany)). The receptacle is filled to approximately half its maximum volume to minimize the risk of spillage.

The patient is seated and nebulization is administered at 8 L/min flow rate of medical air under continuous supervision of the SLT. The face mask is held to the patient’s face by the SLT or by the patient themselves, without using the elastic fixation strings of the mask, so that the mask can be removed quickly if necessary. The patient is instructed to take normal breaths in and out through the mouth. The SLT observes for patient reactions to capsaicin (cough, throat clearing, swallow reflex, discomfort) and, if necessary, assists in immediate discontinuation of the nebulization and removal of the face mask. In our experience, bouts of reflex coughing are triggered after 2–5 breaths in the majority of patients. In one sitting, depending on patient tolerance, up to ten repetitions of up to ten consecutive breaths of inhalation are performed, with short rest periods in between repetitions.

### 5.3. Observations

To document patient reaction and tolerance to capsaicin inhalation, the following observations are made at the beginning and end of each inhalation therapy session:Self-rated patient comfort (feeling thermometer [42]).Oxygen saturation (pulse oximetry).Sputum swallowing frequency per minute (SLT observation).Collection of intraoral saliva status (SLT observation).Quality of voice, e.g., wet voice (SLT observation).

During the inhalation, the SLT notes whether capsaicin triggers a patient reaction, such as coughing, throat clearing or swallow reflex; and notes any changes in voice quality, secretion status, or general wellbeing of the patient.

## 6. Follow-Up and Outcomes

The patient received daily (Monday through Friday) inhalation therapy with nebulized capsaicin for two weeks, from 1 October 2021 to 13 October 2021 (eight sessions in total).

### 6.1. Clinician and Patient-Assessed Outcomes

At the beginning of the very first inhalation therapy session, the SLT observed a wet voice with occasional spontaneous cough, but expectoration remained inadequate. The patient also complained of subjectively perceptible bronchial secretions which he felt unable to clear. During the inhalation with nebulized capsaicin, each round of application triggered a reaction of coughing or throat clearing. During the ninth and tenth applications, it was observed that the patient was able to independently and adequately clear his throat and expectorate bronchial secretions. There was increased drooling of saliva and more fluid saliva. As indicated by the patient’s self-rated comfort level (feeling thermometer pre-inhalation 2/10, post-inhalation 5/10), there was a strong improvement in his general well-being. In conversation, the patient expressed that he felt better after being able to expectorate.

At the beginning of the second inhalation therapy session, the patient reported an improvement in that—for the first time since his admission—he had been able to sleep through the whole night because he was not awakened by coughing attacks at night, and he felt rested and generally more relaxed. The second and all remaining inhalation therapy sessions were similar to the first session. The patient reported a burning sensation in the throat and at the level of the vocal folds during and immediately after the inhalation. Successful throat clearing and expectoration of the bronchial secretion could be observed in every session. Oxygen saturation remained stable throughout all sessions. In general, the patient reported an improvement in well-being despite the irritation from the capsaicin. Clinician and patient-assessed outcomes for all inhalation therapy sessions are presented in Table 2. Nursing staff reported that increased spontaneous throat clearing was observed during supervised mealtimes, especially during lunch about 2 h after inhalation. 

### 6.2. Follow-Up Diagnostics

A follow-up FEES was carried out after one week of inhalation therapy with nebulized capsaicin, on 7 October 2021, showing unchanged silent aspiration (PAS 8) of liquids at IDDSI Level 1. However, the rating of secretions was greatly improved (ROS 1) and there was a moderate response (pharyngeal constriction) to light touch of the scope to the epiglottis. The clearing mechanisms and the throat clearing were found to be more adequate and more sufficient. The pharyngeal and laryngeal mucosa were less inflamed/reddened or completely unremarkable compared to 28 September 2021. Due to these FEES results, daily functional dysphagia therapy was re-commenced while daily inhalation of nebulized capsaicin continued as before.

A final FEES was performed on 14 October 2021, to re-evaluate the status following a further week of inhalation therapy with nebulized capsaicin and additional daily functional dysphagia therapy. There was improvement to be observed as follow:ROS 0.PAS 1 for thickened water IDDSI 2.PAS 1 for thickened water IDDSI 1.PAS 3 for thin water IDDSI 0.Sensory testing: normal response to light touch of the scope to the epiglottis, prompt coughing.Mucous membranes unremarkable.

Consecutive findings from FEES throughout the patient’s rehabilitation stay are summarized in Table 3.

### 6.3. Intervention Adherence and Tolerability 

The patient completed 2 weeks of daily inhalation therapy with nebulized capsaicin. Ten rounds of inhalation per session were well tolerated as demonstrated by the patient’s self-rated level of comfort before and after inhalation (Table 2).

### 6.4. Adverse and Unanticipated Events

Other than a transient burning sensation in the throat and at the level of the vocal folds during and immediately after inhalation, there were no adverse or unanticipated events related to inhalation therapy with nebulized capsaicin. Oxygen saturation remained stable before and after inhalation (Table 2). Visual inspection of the laryngeal structures by FEES on 07 October 2021 and 14 October 2021 (after one and two weeks of inhalation therapy with capsaicin, respectively) did not raise any concerns regarding irritation of the mucosa and vocal folds due to capsaicin.

### 6.5. Remaining Rehabilitation Stay and Discharge from Rehabilitation

In view of the positive findings of FEES on 14 October 2021, inhalation therapy with nebulized capsaicin was discontinued. The patient remained at our rehabilitation center for a further two weeks and received continued functional dysphagia therapy to improve his swallowing function and to reinforce compensation strategies. During this time, the patient’s vital parameters were monitored daily, and there were no signs of aspiration-related pneumonia.

The patient was discharged to his home on 28 October 2021. His FIM score at discharge was 85/91 (motor subscale), 32/35 (cognition subscale), and 117/126 (total score). In his swallowing status, a recommendation was made to continue with modified food textures at IDDSI level 6. Although swallowing of thin liquids with clearing mechanisms such as supraglottic swallowing had been assessed at PAS 3, the patient requested to remain on thickened drinks (IDDSI 1) when he returned home from our facility.

## 7. Discussion

This case report demonstrates the use of inhalation therapy with nebulized capsaicin as an adjunct to inpatient rehabilitation in a patient with oropharyngeal dysphagia post stroke. The aim was to elicit reflex coughs to facilitate secretion clearance from the larynx and lower airways, thereby minimizing aspiration and risk of aspiration-related pneumonia. 

The hypothesis that a therapeutic application of inhaled nebulized capsaicin could be utilized to initiate reflexive coughing and aid the clearance of respiratory secretions from the lower airways has previously been suggested by Kulnik [43]. While this hypothesis refers to a scenario of emergency respiratory physiotherapy requiring nasotracheal suctioning for acute secretion retention [43], our present case report describes a patient with stable respiratory status and therefore a more prophylactic intention for administering nebulized capsaicin.

### 7.1. Scientific Rationale

The scientific rationale for inhalation therapy with nebulized capsaicin as described in this case report is based, firstly, on the neurophysiology of reflex cough [44]; and secondly, on the capsaicin inhalation cough challenge [22,23], which offers a precedent method for intentionally and safely inducing reflex cough in humans.

The main reflexogenic zones for cough are located in the larynx and the tracheobronchial tree, whereby capsaicin acts on transient receptor potential vanilloid 1 (TRPV1) channels [23]. Sensory afferents (Aβ, Aδ and unmyelinated C-fibres) from laryngeal receptors and from the lower airways are conveyed via the superior laryngeal nerve and other vagal afferents to the nucleus tractus solitarii in the brainstem, from where they project further to the pontine and medullary respiratory centers [45]. The efferent pathway of reflex cough leads from the medullary respiratory centers via upper motor neurons to the spinal motor neurons that supply inspiratory and expiratory muscles; via cranial motor neurons to the intrinsic laryngeal muscles that effect coordinated laryngeal abduction and adduction; and via sympathetic and parasympathetic efferents to the smooth airway muscles that effect bronchoconstriction during cough [45,46].

Addington likened the diagnostic use of the inhalation cough challenge to applying a reflex hammer or percussor to the cough reflex, allowing us to gauge the neurophysiological integrity of this crucial airway-protective mechanism [17]. In a similar analogy, the therapeutic application of inhalation with nebulized capsaicin as described in this case report could be likened to ‘jump-starting’ the cough reflex following central neurological impairment caused by stroke. This type of nervous system stimulation has been described as a viable strategy for enhancing recovery and neuroplasticity in post-stroke rehabilitation [46].

### 7.2. Relevant Literature

To our knowledge, inhalation therapy with nebulized capsaicin for the purpose of minimizing aspiration and pneumonia risk in alert and cooperative patients post stroke has not been describe in the literature to date [47]. We are aware of one recently published randomized controlled trial by Wu et al. [48], in which tracheotomized patients post hemorrhagic stroke with moderate to severe impairment of consciousness received one week of four times daily capsaicin nebulization via a tracheostomy mask. This trial reported some positive preliminary findings, with higher sputum output, fewer required suctioning passes and lower clinical pulmonary infection scores in the group that received capsaicin nebulization [48].

Of note, there is a related body of literature, in which capsaicin in different modes of administration is suggested to improve cough and/or swallowing function. This literature includes studies of capsaicin ointment which is applied to the external auditory canal, under the hypothesis that this could stimulate the auricular branch of the vagus nerve (Arnold’s nerve) and improve glottal closure and reflex cough via Arnold’s ear–cough reflex [49]; and studies in which capsaicin is added to ingested drinks or food [50,51] or administered to the oropharynx using a troche [52] or a frozen/cold swab containing capsaicin [53,54]. The latter studies are mainly based on the reasoning that capsaicin improves swallowing function by stimulating and enhancing oropharyngeal and laryngeal sensation [51,54]. In view of this—and bearing in mind the close neuroanatomical and functional relatedness of cough and swallowing—it may therefore be hypothesized that inhalation therapy with nebulized capsaicin could have a similarly beneficial effect also on swallowing in addition to cough function. Such a hypothesis would align with our observations of improving laryngeal sensation and swallowing function in this case report, which coincided with the period of capsaicin inhalation.

### 7.3. Strengths and Limitations of This Report

This report is limited in that case reports offer low-level evidence in medical literature (i.e., Level 4 according to the Oxford Centre for Evidence-Based Medicine) [55], and further studies are required. Strengths of this report include the comprehensive and prospective documentation of patient reaction and tolerance to capsaicin inhalation; and the availability of findings from frequent FEES re-assessments, which were conducted by experienced SLT operators during the course of the patient’s rehabilitation stay. These data demonstrate the safety and tolerability of inhalation therapy with nebulized capsaicin in this mode of application and support the clinical reasoning underpinning the decision to administer capsaicin nebulization in this case.

### 7.4. Conclusions

This case report presents an example of inhalation therapy with nebulized capsaicin in a patient with oropharyngeal dysphagia post stroke, with the aim to facilitate coughing and clearing of secretions and to minimize aspiration and risk of aspiration-related pneumonia. The report demonstrates good patient response to treatment and supports patient safety and tolerability of nebulized capsaicin in this mode of application. Future research should employ a randomized controlled trial design to produce high-level evidence of the effectiveness of nebulized capsaicin in patients with oropharyngeal dysphagia post stroke. The rationale and outcome selection of such a trial should address effectiveness of nebulized capsaicin for prevention of aspiration-related post-stroke pneumonia as well as the effectiveness of nebulized capsaicin in aiding recovery of swallowing function post stroke.

## 8. Patient Perspective

The patient was agreeable to inhalation therapy with nebulized capsaicin when it was first suggested and the purpose explained to him. During inhalation, the patient frequently stated that he noticed pharyngeal irritation as a burning sensation. He also stated that he felt more able to expectorate and had a feeling of ‘free airways’ immediately after the inhalation. The inhalation was administered in the mornings, and the patient reported that this feeling of ‘free airways’ tended to last until noon.

Commenting on his quality of sleep in the night after his very first capsaicin inhalation, the patient reported that, for the first time since his admission, he had been able to sleep through the night without waking from coughing fits. Over the course of inhalation therapy with nebulized capsaicin, the patient remarked that the regular self-evaluation with the feeling thermometer provided him helpful feedback. He felt the self-ratings helped him to reflect on his progress and to see the benefit of the inhalation therapy.

## Figures and Tables

**Figure 1 geriatrics-07-00027-f001:**
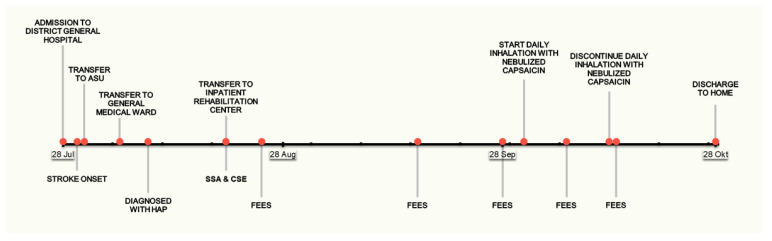
Timeline of the patient’s episode of care. ASU, Acute Stroke Unit; CSE, Clinical Swallowing Evaluation; FEES, Fiberoptic Endoscopic Evaluation of Swallowing; HAP, Hospital Acquired Pneumonia; SSA, Standardized Swallowing Assessment.

**Figure 2 geriatrics-07-00027-f002:**
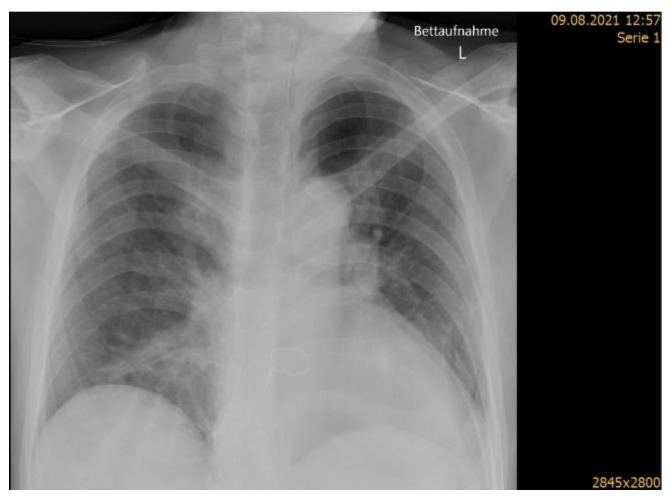
Chest X-ray showing signs of hospital acquired pneumonia (right middle lobe pneumonia).

**Table 1 geriatrics-07-00027-t001:** Assessments of stroke-related impairment and disability during acute hospital admission.

Measure	Admission to DGH	Stroke Onset	Admission to ASU	Discharge from Acute Hospital
NIHSS (points)	n/a	25/42	15/42	1/42
mRS (points)	0	5	5	4

ASU, Acute Stroke Unit; DGH, District General Hospital; NIHSS, National Institutes of Health Stroke Scale [30]; mRS, modified Rankin Scale [31].

**Table 2 geriatrics-07-00027-t002:** Clinician and patient-assessed outcomes for inhalation therapy sessions.

Time Point	Observation	Session 1	Session 2	Session 3	Session 4	Session 5	Session 6	Session 7	Session 8
Pre-inhalation	Self-rated patient comfort *	2	6	5	6	6	5	6	5
	SpO_2_ (%)	94	92	95	92	92	92	92	93
	Sputum swallowing frequency (swallows per minute)	1	0	1	13.1	1	0	1	0
	Intraoral saliva (i.s.) status	Little i.s.	Little i.s.	No excess i.s	Right cheek accumulating i.s	No excess i.s	Foamy saliva at soft palate	Foamy saliva and coated tongue	Unremarkable
	Quality of voice	Wet voice	Normal	Normal	Normal	Normal	Wet voice	Normal	Normal
During inhalation	Coughing	Observed, prompt, throughout							
	Throat clearing	Observed, prompt, throughout							
	Reflex swallow	Observed, prompt, throughout							
	Other observations	Coughed and cleared respiratory secretions	None	Bronchial secretion expectorated, runny nose, mild burning sensation	Runny nose, coughed and cleared secretions, mild burning sensation	Runny nose, moderate burning sensation	Runny nose, continuous throat clearing, wet voice improved	Increasing secretions, adequate throat clearing	Adequate throat clearing, strong burning sensation
Post-inhalation	Self-rated patient comfort	5	4	5	6	6	6	7	5
	SpO_2_ (%)	90	91	91	90	90	91	91	91
	Sputum swallowing frequency (swallows per minute)	2	2	1	1	1	2	1	2
	Intraoral saliva status	Unremarkable	Unremarkable	Unremarkable	Foamy saliva	Unremarkable	Unremarkable	Unremarkable	Unremarkable
	Quality of voice	Wet voice improved	Wet voice, immediately clearing	Adequate clearing	Normal	Normal	Wet voice, adequately clearing	Normal	Normal

* Feeling thermometer, 0–10 (higher ratings indicate higher patient comfort); SpO_2_, oxygen saturation (pulse oximetry).

**Table 3 geriatrics-07-00027-t003:** Fiberoptic Endoscopic Evaluation of Swallowing (FEES) findings during the patient’s rehabilitation stay.

Observation	25 August 2021	16 September 2021	28 September 2021	7 October 2021	14 October 2021
ROS	3	3	2	1	0
PAS at IDDSI level 6	n/a	n/a	5	3	1
PAS at IDDSI level 4	3	3	3	3	1
PAS at IDDSI level 2	5	5	5	5	1
PAS at IDDSI level 1	n/a	n/a	8	8	1
PAS at IDDSI level 0	n/a	n/a	n/a	n/a	3
Sensory test	None	None	Minimal response (pharyngeal constriction), redness	Moderate response (pharyngeal constriction)	Normal response (prompt coughing)
Diet Recommendation	IDDSI 4	IDDSI 4	IDDSI 4	IDDSI 5	IDDSI 6
Fluids Recommendation	IDDSI 2	IDDSI 2	IDDSI 2	IDDSI 2	IDDSI 0
Mealtime supervision (staff to patient ratio)	1:1	1:2	1:2	None	None

IDDSI, International Dysphagia Diet Standardization Initiative; PAS, Penetration Aspiration Scale; ROS, Murray Rating of Secretions scale.

## Data Availability

The data presented in this study are available in this article.

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
