# Peer review of "Inhalation Therapy with Nebulized Capsaicin in a Patient with Oropharyngeal Dysphagia Post Stroke: A Clinical Case Report"

_geriatrics, 2022, doi:10.3390/geriatrics7020027_

Round 1

Reviewer 1 Report

The article is very innovative.
It is valid as a case study, since it lacks a sample.
Table 2 should be edited, and presented in a more aesthetic and clear way, I suggest using abbreviations.

I suggest including a graph to better understand table 3. The graph should be the one relevant to case studies, with different lines that compare the different evaluations.

Reviewer 2 Report

I read with great interest the paper by Pekacka-Egli and colleagues.

This is a very interesting and well-described case report aimed to investigate the efficacy of inhalation therapy with nebulized capsaicin in a Patient With Oropharyngeal Dysphagia Post Stroke.

This study shows good patient response, safety, and tolerability of nebulized capsaicin. This could represent a very interesting and promising therapy that could be administered even in association with swallowing rehab to facilitate coughing and clearing of secretions, and to minimize aspiration and risk of aspiration-related pneumonia post-stroke.

Strength of the paper:

-    The accurate swallowing monitoring through frequent FEES re-assessment,

     which represents the gold standard for detecting dysphagia and aspiration (together

with VFS).

  • The prospective documentation of the case is very accurate, clearly exposed, and complete, with comprehensive clinical observations performed in each inhalation therapy session.
  • The patient’s perspective and capsaicin side effects monitoring are very interesting and well-performed, thus preventing the first objection capsaicin treatment might rice.
  • The paper is very clear, detailed, and also well exposed (I particularly appreciated the “care timeline” and the whole structure of the paper, with great focus on clinical and instrumental evaluation and follow-up, and separated paragraphs on scientific rationale and relevant literature. There positioned, they enrich the issue description and analysis without weighing down the introduction).

-    

Thus, I have just some minor points.

In my opinion, the paper would be considerably improved by reviewing the following minor issues:

  • I strongly agree with the Authors when claiming that “early dysphagia screening and clinical swallowing assessment followed by appropriate clinical management strategies can reduce aspiration and subsequently lower the risk of pneumonia post-stroke”. This is also the reason why post-stroke swallowing impairment incidence ranges from 20 to 80%.

In this regard, I suggest pointing out, with a couple of subsequent lines, two crucial and related aspects.

First, beyond the “early screening”, it also matters the kind of screening (bedside assessment evaluation - BSE) to administer. Actually, many screenings are focused on dysphagia without considering aspiration and vice-versa. This is misleading, at best, because dysphagia may occur without aspiration (please cite Daniels SK, Anderson JA, Willson PC. Valid items for screening dysphagia risk in patients with stroke: a systematic review. Stroke. 2012;43(3):892-7). Conversely, high sensitive BSEs designed to detect also aspiration and tested against FEES (please cite Toscano M et al. Eur J Neurol. 2019 Apr;26(4):596-602) and Martino R et al. Stroke. 2009;40(2):555-61.) are more likely to depict the real situation, thus being more useful to design studies on post-stroke aspiration prevention.

Second, if this is true for aspiration, it is particularly true for silent aspiration, whose detection is failed from most BSEs, and that could be of particular relevance for pneumonia after stroke. In fact, by focusing on the overt sign of aspiration to diagnose post-stroke dysphagia, such as cough or voice change, the silent aspiration would be undiagnosed, with a relevant increase in relative risk of pneumonia and poorer stroke clinical outcome.

Indeed, recent studies reported higher stroke-associated pneumonia:

- in stroke patients who failed high-sensitive screening for dysphagia compared to those who passed the screening (please cite Suntrup-Krueger S et al. Cerebrovasc Dis. 2018;45(3-4):101-8),

- but more important, in stroke patients who passed low-sensitive screening for dysphagia compared to those who passed high-sensitive ones who can detect also silent aspiration. (please cite Jannini TB et al. Neurol Sci. 2022 Feb;43(2):1167-1176.)  

The nutritional intake had initially been managed via NGT. (lines 116-117)

When was it started (i.e. days from the stroke)? Was the patient stable? Was the NGT feeding regimen preceded by a parenteral one in the acute phase?

Please add these data. This is important because regurgitation and aspiration may be significantly higher in stroke patients who undergo an NGT feeding regimen. The presence of a high gastric residual volume, coupled with stroke-induced deglutition impairments, may easily lead to aspiration of gastric content in the lungs.

Moreover, both dislodgement of the tube and the repetition of the positioning procedure are risk factors for developing pneumonia.

Why did on 16 and 21 of September you performed the 3-oz screening (which has relatively low sensitivity, especially for silent aspiration, see above) instead of those more sensitive who was administered in August?
